# Genetic Characteristics of Measles Viruses Isolated in Taiwan between 2015 and 2020

**DOI:** 10.3390/v15010211

**Published:** 2023-01-12

**Authors:** Wen-Yueh Cheng, Bao-Shen Chen, Hsiao-Chi Wang, Ming-Tsan Liu

**Affiliations:** Center for Research, Diagnostic and Vaccine Development, Centers for Disease Control, Ministry of Health and Welfare, Taipei 11561, Taiwan

**Keywords:** measles, genotype, elimination

## Abstract

A genetic analysis of circulating measles virus (MeV) provides strong evidence of an interruption in endemic measles and supports the elimination status of this disease. This study investigated 219 MeVs isolated between 2015 and 2020. Based on the 450 nucleotide sequences of the nucleoprotein gene (N-450), three genotypes of the H1, D8 and B3 with 8, 18 and 6 different N-450 sequences, respectively, were identified. The H1 genotype virus has not circulated in Taiwan since 2017, and the D8 and B3 genotype MeVs became dominant between 2018 and 2019. Different D8 genotype variants were imported from neighboring countries, and the majority of MeV variants were detected only for a short period. However, MVs/Gir Somnath.IND/42.16[D8], a named strain designated by the World Health Organization (WHO), was detected over 2 years. To explore whether the endemic transmission of measles has been underestimated, another sequence window of the hypervariable, noncoding regions between the matrix (M) and fusion (F) genes (MF-NCR) was introduced to clarify the transmission chain. From the chronological sequence analysis of MeVs with N-450 and MF-NCR sequence windows, no endemic MeV variants lasted over 4 weeks, providing strong evidence to support the contention that Taiwan has reached the status for measles elimination.

## 1. Introduction

Measles is one of the most contagious diseases, with an estimated reproductive number of 12–18 in a completely susceptible population [1]. During the pre-vaccine era, it was one of the leading causes of death in children younger than 5 years old [2]. Measles is also feasible to eradicate by meeting three biological criteria, including humans being the only pathogen reservoir, having an effective vaccine available at reasonable cost and having accurate diagnostic tests [3]. Countries in all six WHO regions adopted goals for measles elimination by or before 2020 [4]. Measles elimination is defined as the absence of endemic measles virus transmission in a region or other defined geographical area for ≥12 months in the presence of a well-performing surveillance system [5]. Moreover, the verification of elimination requires evidence that transmission has been interrupted for at least 36 months [6]. To verify the interruption of endemic measles virus transmission, viral sequence analysis combined with case investigations are needed to document the absence of endemic genotypes. In Taiwan, the measles cases decreased since 2015, and were the lowest in 2017 with only 6 cases confirmed, but it rebounded to 140 cases in 2019 as the worldwide trend [4]. The coverage rate of the first dose measles, mumps and rubella (MMR) vaccine in children aged 12 months old reached over 98% between 2015 and 2020, and the second dose MMR that was given to children before entrance to elementary school was improved from 93.42% in 2015 to 96.77% in 2020. Among 230 measles cases detected between 2015 and 2020, 7.8% were from children younger than 1 year old, 3.9% for children aged 1–6 years old, 1.3% for school-age children (7–17 years old), 77.8% for adults aged 18–40 years old and 9.2% for adults aged over 40 years old. Nearly 74% cases were from northern Taiwan, and those from central and southern Taiwan accounted for 15.5% and 10.5%, respectively. In total, 24 MeV genotypes were identified, and the number of genotypes detected per year decreased from 13 in 2002 to three in 2020, a sign of progress toward elimination [4]. To discriminate between the most frequently detected measles strain of genotypes D8 and B3, a “named strain” based on the identity of N-450 sequences was introduced to better reflect currently circulating viruses. Even so, when faced with a repeated importation or outbreak with the same recurring N-450 variants, the use of an extended sequencing window, the hypervariable, noncoding regions between the matrix (M) and fusion (F) gene (MF-NCR) as recommended by the WHO Global Measles and Rubella Laboratory Network (GMRLN) [7] was first tied to the genotypes for D8 and B3 MeVs identified between 2015 and 2020. To reach the measles elimination goal, case-based surveillance of every reported measles case either in epidemiological investigation or laboratory confirmation has been ongoing in Taiwan since 2000 [8]. Laboratory surveillance data from before 2014 have been described before [9,10,11], and updated laboratory data on genetic characteristics explored in view of the N-450 “named strain” and MF-NCR are presented in this study.

## 2. Materials and Methods

### 2.1. Collection of Clinical Samples

Throat swabs and urine were collected from suspected measles cases that were reported to the Taiwan Centers for Disease Control (Taiwan CDC) for laboratory confirmation of measles disease.

### 2.2. Viral RNA Detection

Total RNA was extracted directly from the throat swabs and urine with a TANBead Nucleic Acid Extraction Kit on a Smart LabAssist instrument (Taiwan Advanced Nanotech Inc., Taoyuan, Taiwan) according to the manufacturer’s instructions. One-step real-time RT-PCR (RT-rPCR) reactions were performed with Roche LightCycler Multiplex RNA Virus Master Mix (Roche Diagnostics GmbH, Mannheim, Germany). The RT-rPCR mixtures (total volume, 20 µL) were incubated at 50 °C for 10 min (RT step) and 95 °C for 30 s (denaturation) and subjected to 45 cycles of amplification (95 °C for 5 s and 60 °C for 20 s) on the Roche LightCycler 480 instrument using a primer and probe set designed to MeV phosphoprotein gene with forward primer 1905F, reverse primer 2030R and probe 1967P (Table 1). The RT-rPCR result was defined as positive if the cycle threshold value of RT-rPCR was less 40.

### 2.3. Virus Isolation, Genotyping and Extended Window RT-PCR

When throat and urine samples were confirmed positively for MeV by RT-rPCR assay, the samples were inoculated into Vero/hSLAM cells-a cell line developed by Dr. Yangi and colleagues at Kyushu University, Fukuoka, Japan, for further isolation of the MeVs [12]. The culture medium was supplemented with Geneticin to retain SLAM (signaling lymphocyte-activation molecule) expression [13].

Genotyping experiments were carried out with specimens of MeV confirmed cases with positive RT-rPCR results by using nested RT-PCR assays. The nested RT-PCR was performed first using a one-step RT-PCR kit (Qiagen, Hilden, Germany) with forward primer MV59 and reverse primer MV64 as described before [8] and a nested PCR was performed on the resulting PCR product using SapphireAmp^®^ Fast PCR Master Mix (TAKARA BIO, Shiga, Japan) with primers Me214 and Me216 [10].

For extended window sequencing, two overlapping segments encompassing the MF-NCR region were amplified with a one-step RT-PCR kit (Qiagen, Hilden, Germany) with primer set 1 of forward primer 4200F and reverse primer 4869R and primer set 2 of forward primer 4801F and reverse primer 5609R. Then, hemi-nested PCR was performed using this PCR product as a template and forward primer set 1 and primer set 2 were replaced with 4212F and 4811F, respectively, with Dream Taq PCR Master Mix (Thermo Fisher Scientific Baltics, Vilnius, Lithuania) as a reagent. The detailed sequences of the primer sets are listed in Table 1.

### 2.4. Sequencing and Phylogenetic Analysis

Sequencing reactions were performed with an ABI BigDye Terminator V3.1 cycle sequencing kit (Applied Biosystems, Foster City, CA, USA) according to the manufacturer’s instructions. The forward and reverse sequencing primers used were Me 216 and Me 214 for N-450 and 4212F/4869R, 4811F/5609R for segment 1 and segment 2 of MF-NCR, respectively. The MeV genotypes were determined by phylogenetic analysis N-450 together with WHO-designated reference sequences for each genotype. A phylogenetic tree was constructed with MEGA software version 11.0 [14] based on neighbor-joining methods using 1000 bootstrap replicates. The sequences generated in this study are available under the following NCBI accession numbers: N-450–OP680022 to OP680240; and MF NCR–OP680241 to OP680397.

### 2.5. Ethic Statement

Measles is a notifiable disease in Taiwan, and clinical specimens of suspected cases must be collected and tested for the measles virus according to the Communicable Disease Control Act. This study did not involve any activities that were reviewed prospectively by an ethics committee, and informed consent from suspected cases was exempted. This study was approved for publication by the Taiwan CDC.

## 3. Results

In total, 1979 suspected measles cases were reported to the Taiwan CDC between 2015 and 2020, with fewer than 100 reported cases in 2017 and 982 in 2019 (Table 2). Among the laboratory-confirmed cases (*n* = 230), 223 were MeV RT-rPCR positive, and 7 were RT-rPCR negative but were diagnosed as measles IgM positive. The genotypes with available N-450 sequences (*n* = 219) were composed of H1 (*n* = 35), D8 (*n* = 146) and B3 (*n* = 38) (Table 2). The last H1 genotype was detected in 2017, the D8 became the most frequently detected genotype since 2017, and the B3 genotype increased from 2017 and peaked in 2019 (Table 2, Appendix A).

### 3.1. Analysis According to N-450 Sequences

#### 3.1.1. Genotype H1

Genotype H1 MeVs detected in Taiwan consisted of eight different sequence variants (H1-seq 1 to H1-seq 8 (H1 seq 1–8)) (Table 3, Appendix A). A phylogenetic analysis was performed using sequences consisting of the H1 seq 1–8 variant, a WHO reference strain of genotype H1 (MVi/Hunan.CHN/0.93/7[H1]), and five WHO-named strains. In addition, four overseas genotype H1 MeVs with identical N-450 sequences to one or another of the H1 seq 1–8 variants were included for comparison (Figure 1). The H1-seq 1 (*n* = 24) variant had the identical N-450 sequence to WHO-named strain MVs/Hong Kong.CHN/49.12[H1] (Table 3). This sequence was detected between the 12th and 27th weeks in 2015 from multiple importations from China and was associated with a domestic transmission cluster in a tax-free shop in 2015 (Table 4, Epi-link 1). The H1-seq 3 variant (*n* = 2) had the identical N-450 sequence to WHO-named strain MVs/Hong Kong.CHN/42.11[H1], one sequence linked to a case with travel history to China, and the other was from a domestic case. H1-seq 5 (*n* = 1) had an identical N-450 sequence to MVs/Aichi. JPN/20.14/1 (accession number AB968373) and MVs/Osaka C. JPN/25.14 (accession number LC002658) and was linked to importation from Vietnam. The H1-seq 7 (*n* = 4) variant was linked to two independent importations from China and two were from domestic transmission, had a sequence identical to MVs/Berlin.DEU/39.16/2 (accession number KY056247), H1-seq 8 (*n* = 1) had an identical N-450 sequence to MVi/Anhui.CHN/17.17/4 (accession number MH979988), and was linked to importation from China. H1-seq 2 (*n* = 1), H1-seq 4 (*n* = 1) and H1-seq 6 (*n* = 1) variants could not be matched to identical N-450 sequences deposited in GenBank. H1-seq 2 was identified from a case with epidemiological linkage (epi-link) to one measles transmission chain (epi-link 1, Appendix A), and H1-seq 4 and H1-seq 6 were linked to importation from China.

#### 3.1.2. Genotype B3

The genotype B3 MeVs detected in Taiwan consisted of six different sequence variants (B3-seq 1 to B3-seq 6 (B3 seq 1–6) variants, Appendix A). A phylogenetic analysis was performed using sequences consisting of the B3 seq 1–6 variants, two WHO reference strains of genotype B3 (MVi/New York.USA/0.94[B3], MVi/Ibadan.NGA/0.97/1[B3]), and 21 WHO-named strains. In addition, five overseas B3 strains with identical N-450 sequences to one or another of the B3 seq 1–6 variants were included for comparison (Figure 1). The B3-seq 4 (*n* = 25) variant had the identical N-450 sequence to WHO-named strain MVi/Marikina City/PHL/10.18[B3]; this sequence was first detected during the 6th week in 2019 and was linked to importation from the Philippines, another detected during the 13th week in 2019 was related to an imported case from China and three sequences were from cases with travel history to European countries (Sweden, Norway, Denmark, Finland, Iceland and England) between the 13th and 15th week in 2019, and the other 19 sequences detected between the 6th and 20th week in 2019 were from cases without any travel history. The B3-seq 2 (*n* = 8) variant had an identical N-450 sequence to WHO-named strain MVi/Gombak.MYS/40.15[B3]. Three sequences were detected in 2018, and the other five were detected in 2019. All cases except one had a travel history to the Philippines. The B3-seq 1 (*n* = 1) variant had an identical N-450 sequence to WHO-named strain MVi/Gombak.MYS/44.16[B3], which was detected during the 1st week in 2018 from a case with an unidentified source of infection. The B3-seq 3(*n* = 1) variant was linked to importation from the Philippines and had an identical sequence to MVs/Stockholm.SWE/10.19[B3] (accession number MK633031). The B3-seq 5 (*n* = 1) variant was isolated from a case who had a travel history to the USA and China, with an identical sequence to MVs/British Columbia.CAN/4.19/[B3] (accession number MK386953), MVs/California.USA/7.19/[B3] (accession number MT789788), MVs/California.USA/9.19/[B3] (accession number MT789789) and MVs/California.USA/9.19/2[B3] (accession number MT789790). The other variant B3-seq 6 (*n* = 2) was from cases with an unknown source of infection and could not match with identical N-450 deposited sequences in GenBank.

#### 3.1.3. Genotype D8

Genotype D8 MeVs detected in Taiwan between 2016 and 2020 could be classified into 18 different sequence variants (D8-seq 1 to D8-seq 18 (D8 seq 1–18) variants, Appendix A). A phylogenetic analysis was performed using sequences consisting of the D8 seq 1–18 variant, WHO reference strain of genotype D8 (MVi/Manchester.GBR/30.94 [D8]), and 22 WHO-named strains. In addition, eight overseas D8 strains with identical N-450 sequences to one or another of the D8 seq 1–18 variants were included for comparison (Figure 2). Eight of the D8 seq 1–18 variants had identical sequences to the known WHO-named strain MeVs (Table 3). The D8-seq 3 (*n* = 4) variant identical to WHO-named strain MVs/Osaka,JPN/29.15[D8] was associated with imported cases from Thailand and Japan. The D8-seq 4 (*n* = 1) variant identical to WHO-named strain MVs/Victoria.AUS/6.11[D8] was isolated in 2016 from a captain with a scheduled flying route between India, China, Hawaii and Singapore. Regarding the D8-seq 6 (*n* = 3) variant identical to WHO-named strain MVi/Hulu Langat.MYS/26.11[D8], one sequence detected in 2017 was linked to importation from Europe countries, and the other two sequences detected in 2018 and 2019 were from cases without travel history abroad. The D8-seq 8 (*n* = 90) variant identical to WHO-named strain MVs/Gir Somnath.IND/42.16[D8] had begun to be detected in the 13th week in 2018, and the last one in the 1st week in 2020. The epidemiological investigation suggested that the possible origins included multiple importation events from surrounding Asian countries such as Vietnam, Thailand, Indonesia, Japan, Korea, the Philippines and even farther into Belgium and New Zealand (Appendix A), indicating this D8-seq 8 (identical to MVs/Gir Somnath.IND/42.16) variant was distributed widely and globally. Additionally, the D8-seq 8 variant also caused a domestic transmission cluster at the international airport in 2018 (Epi-link 5, Table 4). The D8-seq 11 (*n* = 4) variant identical to the WHO-named strain MVs/Samut.Sakhon.THA/49.16[D8]: two sequences each from 2018 and 2019 were all linked to cases with travel history to Thailand. The D8-seq 12 (*n* = 1) variant identical to WHO-named strain MVs/Herborn.DEU/05.17 [D8] was linked to importation from England. The D8-seq 13 (*n* = 6) variant identical to WHO-named strain MVs/Samut Sakhon.THA/8.18[D8]: one sequence in 2018 was linked to importation from Cambodia, the other five sequences in 2019 were linked to importation from Thailand. The D8-seq 14 (*n* = 4) variant identical to WHO-named strain MVs/Dagon Seikkan.MMR/5.18[D8] was linked to importation from Thailand and Myanmar. Four N-450 variants of D8-seq 1, 5, 10 and 17 were not identical to the known WHO D8-named strain but could match with identical sequences in GenBank: the D8-seq 1 (*n* = 1) variant in 2016 with an unknown source of infection was identical to MVs/Andernach.DEU/16.16[D8] (accession number KX377943); the D8-seq 5 (*n* = 1) variant identical to MVs/Tucuman.ARG/15.17/30[D8] (accession number MF092792) was linked to importation from Indonesia; the D8-seq 10 (*n* = 1) variant was from a domestic case, and identical to MVs/Maldah.IND/11.17/12[D8] (accession number MG652493); the D8 seq 17 variant (*n* = 6), three of which were associated with importation from Thailand and identical to MVs/California.USA/38.19 (accession number MT989845), MVs/Phatthalung.THA/22.19 (accession number MT555111) and MVs/KobeC.JPN/48.19/19MR90 (accession number LC521318). The other six D8 variants (seq 2, seq 7, seq 9, seq 15, seq 16 and seq 18) could not match with any identical N-450 sequences in GenBank. The D8-seq 2 (*n* = 1) variant was linked to importation from India; the D8-seq 7 (*n* = 1) variant was linked to importation from Thailand. The D8 seq- 9 (*n* = 3) variant and D8- seq16 (*n* = 1) variant detected in 2018 and 2019, respectively, were all linked to importation from Indonesia; the D8-seq 15 (*n* = 12) variant that consisted of 10 sequences from transmission chain Epi-link 13, a nosocomial infection event, was linked to importation from China, and the other two sequences detected from the same household classified as transmission chain Epi-link 17 were linked to importation from Japan. The D8-seq 18 (*n* = 6) variant that included one sequence was linked to importation from Thailand and the other five sequences were linked to importation from Cambodia (Appendix A).

### 3.2. Analysis according to MF-NCR Sequences

Genotypes B3 and D8 were two widely distributed genotypes identified in 2018 and 2019 from countries all over the world [15,16,17,18,19]. When multiple importations with identical N-450 sequences made it difficult to clarify the transmission chain, the use of N450 for molecular epidemiological analysis was not recommended but should rather be used to confirm or rule out epidemiological linkage. Here, we tried the new sequence windows of the MF-NCR region, another hypervariable sequence segment of measles virus, to further analyze the variants of genotypes D8 and B3 MeVs.

#### 3.2.1. MF-NCR Analysis for Genotype B3

A total of 30 of 38 genotype B3 viruses were re-analyzed with an MF-NCR sequence window (Table 2) and re-classified into 11 variants (B3-V1 to V11). Among the 11 MF NCR variants, 2 variants (B3-V5 and B3-V7) could match with identical sequences deposited in GenBank. The B3-V5 (*n* = 1) variant identical to MVs/Hong Kong.CHN/10.19[B3] (accession number MN702462) was isolated at the 6th week in 2019 from a case with travel history to the Philippines. The B3-V7 (*n* = 16) variant was identical to MVs/HongKong.CHN/11.19/3[B3] (accession number MN702465) and three sequences were linked to cases with travel histories to European countries (Sweden, Norway, Denmark, Finland and Iceland, Hong Kong and Japan). The other nine MF- NCR variants could not be matched to any identical sequences in GenBank. Most of the MF-NCR variants of genotype B3 were linked to importation from the Philippines (B3-V2, V3, V4, V5, V9 and V11), one variant was linked to a case with travel history to the USA and China (B3-V6) and the other three variants (B3-V1, V8 and V10) were from domestic cases with unknown sources of infection. A phylogenetic analysis was performed using a dataset consisting of the B3 V1-V11 variants, and MF NCR sequences identical to the B3-V5 and B3-V7 variants (Figure 3).

#### 3.2.2. MF-NCR Analysis for Genotype D8

A total of 127 of 146 genotype B3 viruses were re-analyzed with an MF-NCR sequence window (Table 2) and re-classified into 39 variants (D8-V1 to D8-V39). Among the 39 MF-NCR variants, 5 variants (D8-V3, V10, V12, V20 and V33) could match with identical sequences in GenBank. The D8-V3 (*n* = 1) variant identical to MVs/Prahova.ROU/18.17/2[D8] (accession number T789684) was linked to importation from Thailand; the D8-V10 (*n* = 17) variant consisted of sequences from different N-450 variants including D8-seq 8 (*n* = 9), D8-seq 15 (*n* = 3) and D8-seq 18 (*n* = 5). Combination analysis of N-450 and MF-NCR sequences indicated that D8-seq 8/V10 variant (*n* = 9) was isolated during the period of 3rd–26th week in 2019 from five cases with travel histories to Vietnam, Indonesia, Belgium and China, and four cases with unknown sources of infection and the D8-seq 15/V10 (*n* = 3) variant was isolated from the 13th to 15th week in 2019 from one case with travel history to China and two cases with unknown sources of infection. The D8-seq 18/V10 (*n* = 5) variant was isolated from the 36th to 42nd week in 2019 and one was linked to importation from Thailand and the other four sequences were linked to importation from Cambodia. Identical sequences to D8-V10 were also reported during the 22nd–28th weeks in 2019 from the USA under accession numbers MT789833, MT789838, MT789839, MT789844, MT789848 and MT789849 deposited in GenBank and 13 sequences reported during the 4th-17th weeks in 2019 from Korea under accession numbers MN863753, MN863762, MN863764, MN863765, MN863767, MN863771, MN863774, MN763775, MN863778, MN863779, MN863782, MN863792 and MN863794 (Figure 3). The D8-V12 (*n* = 48) variant included sequences isolated in cases with three different N-450 variants, including D8-seq 8 (*n* = 38), D8-seq 14 (*n* = 3) and D8-seq 15 (*n* = 7). The D8-seq 8/V12 variant (*n* = 38) was first detected during the 14th week in 2018 and the last variant was detected during the 30th week in 2019 from multiple importations from cases with travel histories to Thailand, Vietnam, Japan and Korea and caused domestic transmission, including cases in Epi-links 5, 10 and 11. The D8-seq 14/V12 (*n* = 3) variant isolated during the 53rd week in 2018 was from a source-untraceable case and the other two sequences detected during the 12th and 18th weeks in 2019 were linked to importation from Myanmar and Thailand, respectively. The D8-seq 15/V12 variant (*n* = 7) was isolated during the 13th–15th week period in 2019 from cases with travel histories to Japan or cases from domestic transmission and identical sequences to D8-V12 were also reported during the 12th–18th week period in 2019 from Australia, Luxembourg and the USA under accession numbers MN545787, MW132155, MW132156, MT789825, MT789829 and MT787847 deposited in GenBank and during the 2nd–21st weeks in 2019 from Korea under accession numbers MN863738, MN863741, MN863756, MN863759, MN863760, MN863763,and MN863789 deposited in GenBank. The D8-V20 variant (*n* = 1) isolated in Taiwan during the 36th week in 2019 was linked to importation from New Zealand and identical sequences were deposited in GenBank under accession numbers MN545809 and MT789842, which had been reported from the 30th week and 34th week in 2019 from the USA and Australia, respectively. The D8-V33 (*n* = 5) variant consisted of sequences from two N-450 variants. The D8-seq 13/V33 (*n* = 3) variant included one sequence isolated in 2018 that was linked to importation from Cambodia and the other two sequences isolated during the 14th–15th week in 2019 were linked to importation from Thailand. The D8-seq 17/V33 (*n* = 2) variant included one sequence isolated during the 36th week in 2019 from a case with an unknown source of infection and the other sequence isolated during the 52nd week in 2019 was linked to importation from Thailand and identical sequences to D8-V33 were reported from the USA during the 38th week in 2019 (accession number MT789845) and Korea during the 9th week in 2019 (accession number MN863758). The other 34 MF NCR variants could not match to any identical sequences deposited in GenBank. A phylogenetic analysis was performed using a dataset consisting of the MF NCR sequences of D8-V1 to D8-V39 variants and identical MF-NCR sequences to D8-V3, V10, V12, V20 and V33 variants deposited in GenBank (Figure 3).

## 4. Discussion

Numbers of global MeV genotypes have decreased gradually because of the implementation of the measles elimination program; MeV genotypes detected during ongoing transmission decreased from six (B3, D4, D8, D9, G3 and H1) in 2016 to four (B3, D4, D8, and H1) in 2018. In 2018, genotypes B3 and D8 accounted for 95% of the reported sequences [20]. Measles genetic surveillance in Taiwan also showed the same trend. Genotypes H1, D8 and B3 were identified between 2015 and 2020, and since 2018, only genotypes B3 and D8 have been reported. In addition, the N-450 sequences were less divergent than MeVs detected before 2017. During the 2019 measles epidemic, multiple importations either from the same or different countries caused secondary transmission within a certain period. They shared the same N-450 sequences and this meant that it was difficult to track the transmission chain through traditional epidemiological investigations, so another hypervariable sequence window for MF NCR was first attempted to investigate the correlation between these cases.

Through the epidemiological investigation, 26 epidemiological link (Epi-link) transmission chains (Epi-link 1–26) were identified in 219 MeVs isolated between 2015 and 2020 (Table 4). The case number of each transmission chain included a minimum of 2 cases and a maximum of 21 cases, and Epi-links 1, 5 and 15 accounted for more than 10 cases. Epi-link 1 included 16 cases at a tax-free shop that occurred in 2015. The majority of MeVs (*n* = 15) were classified as H1-seq 1 variants according to the N-450 sequence. One MeV (*n* = 1) was classified as an H1-seq2 variant. The source of Epi-link 1 was untraceable, and the movement of international travelers might be the origin of this transmission chain. Epi-link 5 included 21 cases caused by genotype D8 MeV and associated with an international airport transmission in 2018, an index case that traveled to Thailand approximately two weeks prior, continuing his travel to Okinawa, flying to Okinawa, with an incubation period before onset, to cause secondary transmission at the international airport passport control gate, duty-free shop and aircraft. Infecting cabin crews and international travelers at the secondary transmission chain caused tertiary transmission to other cabin crews, international travelers, ground staff and even the patients in two hospitals [21]. The identical N-450 (D8-seq 8) and MF-NCR (D8-V12) variant sequences supported the result of epidemiological investigation. Epi-link 15 included 18 confirmed cases originating from an index case with a travel history abroad during the process of seeking medical treatment after returning home, causing secondary transmission in the dining area and hospital, followed by a tertiary transmission that expended to a science and technology park. Epi-link 13 and Epi-link 23 included 10 cases each, which were associated with nosocomial transmission at two hospitals indicated by epidemiological investigation.

Mutation frequencies in RNA viruses typically range between 10^−6^ and 10^−3^ per site per replication. In the field, MeV has been shown to maintain high levels of genetic stability, particularly in outbreak settings. MeV was estimated to have a mutation rate of 1.8 × 10^−6^ to 9 × 10^−5^ per base per replication [22,23], and the standard genome length of measles virus was 15,894 bases; in other words, the genomic mutation rate was 0.028 to 1.43 per replication. When comparing sequences of the N-450 and MF-NCRs, one sequence deviation might imply a different transmission chain, when following this rule, that is aimed at N-450 sequences only. The chain for epi-link 1 consists of 2 H1 N-450 variants (H1-seq 1 and H1-seq 2), and the chain for epi-link 15 consists of 2 B3-N450 variants (B3-seq 4 and B3-seq 6), which might indicate the existence of more than one infection source (Table 4). If the target sequence window was set at the MF-NCR sequence window, 5 of the 22 epi-link chains with available sequences had two (Epi-link 9 with D8-V11 and D8-V17, Epi-link 12 with D8-V12 and D8-V34, Epi-link15 with B3-V7 and B3-V8, Epi-link 18 with B3-V7 and B3-V10) or three variants (Epi-link 13 with D8-V10, D8-V12 and D8-V35) in these chains. The epidemiological link by traditional investigation might find common exposure locations, such as hospitals and dining areas, and the characteristic of the 2019 measles outbreak was multiple transmission from abroad then induced secondary or even tertiary transmission, and it was difficult to follow the exact transmission chain to the source case. When comparing the results of traditional epidemiological investigations and molecular epidemiology, with 77% (17/22) to 92% (24/26) matching according to the MF-NCR or N-450 window, not all confirmed cases could be traced to the origin of the source case by traditional investigation, and not all confirmed cases could successfully yield sequence data from the target window. We should make good use of these two-surveillance data to discriminate better among each case at the measles elimination phase. Among the transmission chain associated with more than 10 cases (epi-link 1, 5, 13, 15 and 23), the period of epidemic lasted no more than 6 weeks (2–6 weeks). Source-untraceable cases accounted for 23.7% (52/219, Appendix A), including 29 cases from seven source-untraceable Epi-link chains (Table 4). The chronological order of MeV variants tracked by combination of N-450 and MF-NCR sequences provided further evidence that no continuing measles circulating transmission occurred in Taiwan between 2015 and 2020.

Different MeV variants that were classified according to the N-450 sequence window might have had identical MF-NCR sequences, and vice versa. For example, MeVs from N-450 variants of D8-seq 8, D8-seq 15 and D8-seq 18 might had identical MF-NCR sequence to D8-V10 variant, and MF NCR variant D8-V9 to D8-V27 might originated from MeVs that had identical N-450 sequence to D8-seq 8 variant. According to the phylogenetic tree of MF-NCR, D8-V10 and D8-V12 were included in the D8-1 sub lineage and the D8-V33 variant consisted of D8-seq 13 and D8-seq 17 that belonged to the D8-2 sub lineage (Figure 3). When viewing the phylogenetic tree of N-450 (Figure 2) indicated that D8-seq 8, D8-seq 14, D8-seq 15 and D8-seq 18 variants were located in the D8-N1 sub lineage and D8-seq 13 and D8-seq 17 were located in the D8-N2 sub lineage; the same trend was identified in genotype B3 MeV detected in Taiwan with B3-V7 variant in B3-1 sub lineage, and included MeVs from B3-seq 4 and B3-seq 6 variants (Figure 3). When viewing the phylogenetic tree of N-450 (Figure 2) indicated the MeVs of B3-seq 4 and B3-seq 6 cluster in the same sub lineage of B3-N1, the same as genotype D4 MeVs reported by the Spanish [24]. The further extension of this window to the whole genome of these highly similar MeVs might help investigators to understand the evolution of MeVs.

## 5. Conclusions

The MeV sequence window targeting the N-450 has been built in Taiwan since 1992, and up to 10 genotypes (B3, D3, D4, D5, D8, D9, G2, G3, H1 and H2) were detected through 2020. A new sequence window of MF-NCR offers a higher resolution compared to N-450 for genotype D8 MeVs (39 variants vs. 18 variants) and genotype B3 MeVs (11 variants vs. 6 variants). It is highly recommended that MF-NCR sequencing is applied as an extra tool for molecular epidemiology and to define chains of transmission. Because the same MeV variants that were classified according to the N-450 sequence window might have different MF-NCR sequences, and vice versa, a deeper analysis of N-450, MF-NCR sequencing, and the whole genome combined with epidemic investigation is ongoing. The data of genotypes according to two sequence windows provide evidence that measles elimination has been reached in Taiwan.

## Figures and Tables

**Figure 1 viruses-15-00211-f001:**
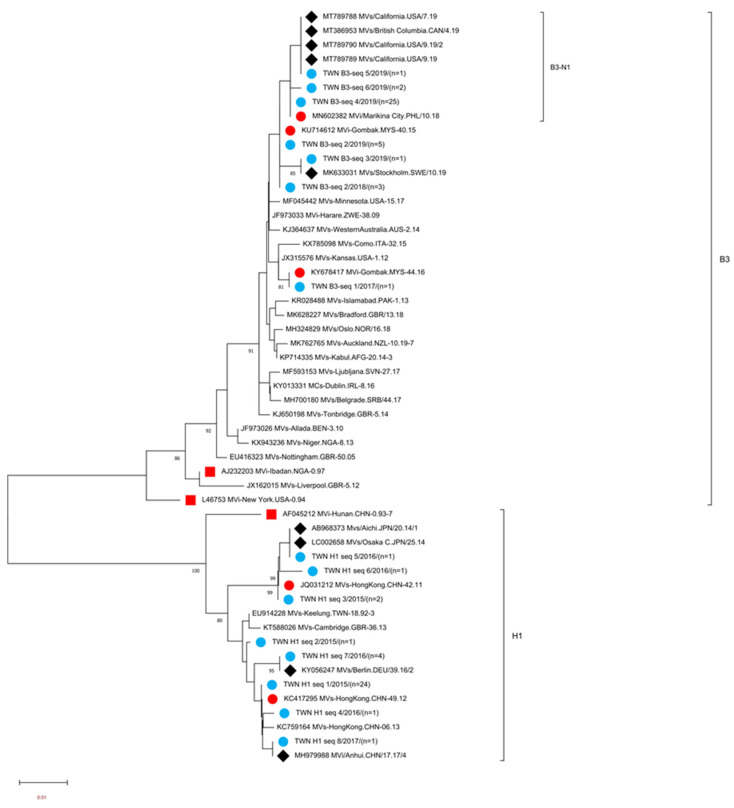
Phylogenetic tree of the genotype B3 and H1 MeV strains. The phylogenetic tree was constructed using the neighbor-joining method based on the N-450 nucleotide sequences. Circles colored in blue indicate strains detected in Taiwan between 2015 and 2020. Circles colored in red indicate the N-450 sequences of WHO-named strains identical to MeV variants in Taiwan. The square colored in red indicates WHO reference MV strains. Diamonds colored in black indicate the N-450 sequences of variants in GenBank identical to MeV variants in Taiwan. Sequences without additional labels were other WHO-named strain viruses with genotypes B3 and H1. The number (n) of each MeV strain with an identical N-450 sequence is shown. Genotype B3 sublineage B3-N1 tentatively proposed in this study is indicated.

**Figure 2 viruses-15-00211-f002:**
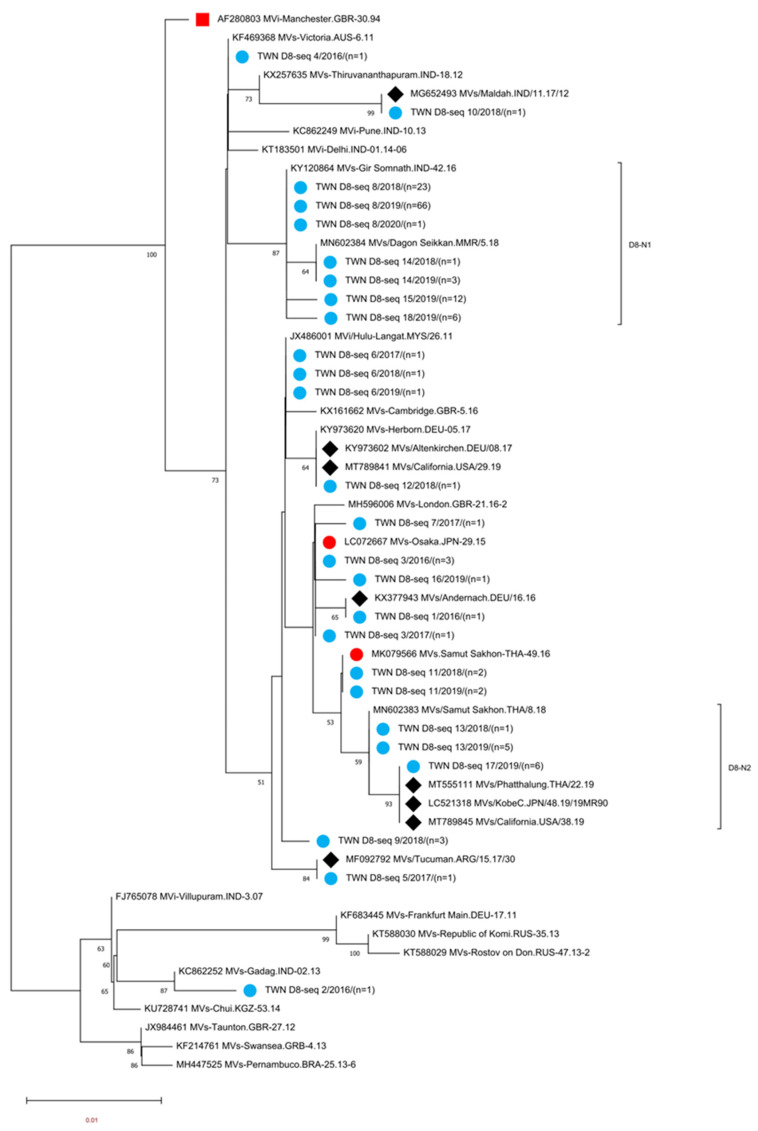
Phylogenetic tree of the genotype D8 MeV strains. The analysis and labeling methods of the MeV strains are described in the legend of Figure 1. Genotype D8 sublineage D8-N1 and D8-N2 as tentatively proposed in this study are indicated.

**Figure 3 viruses-15-00211-f003:**
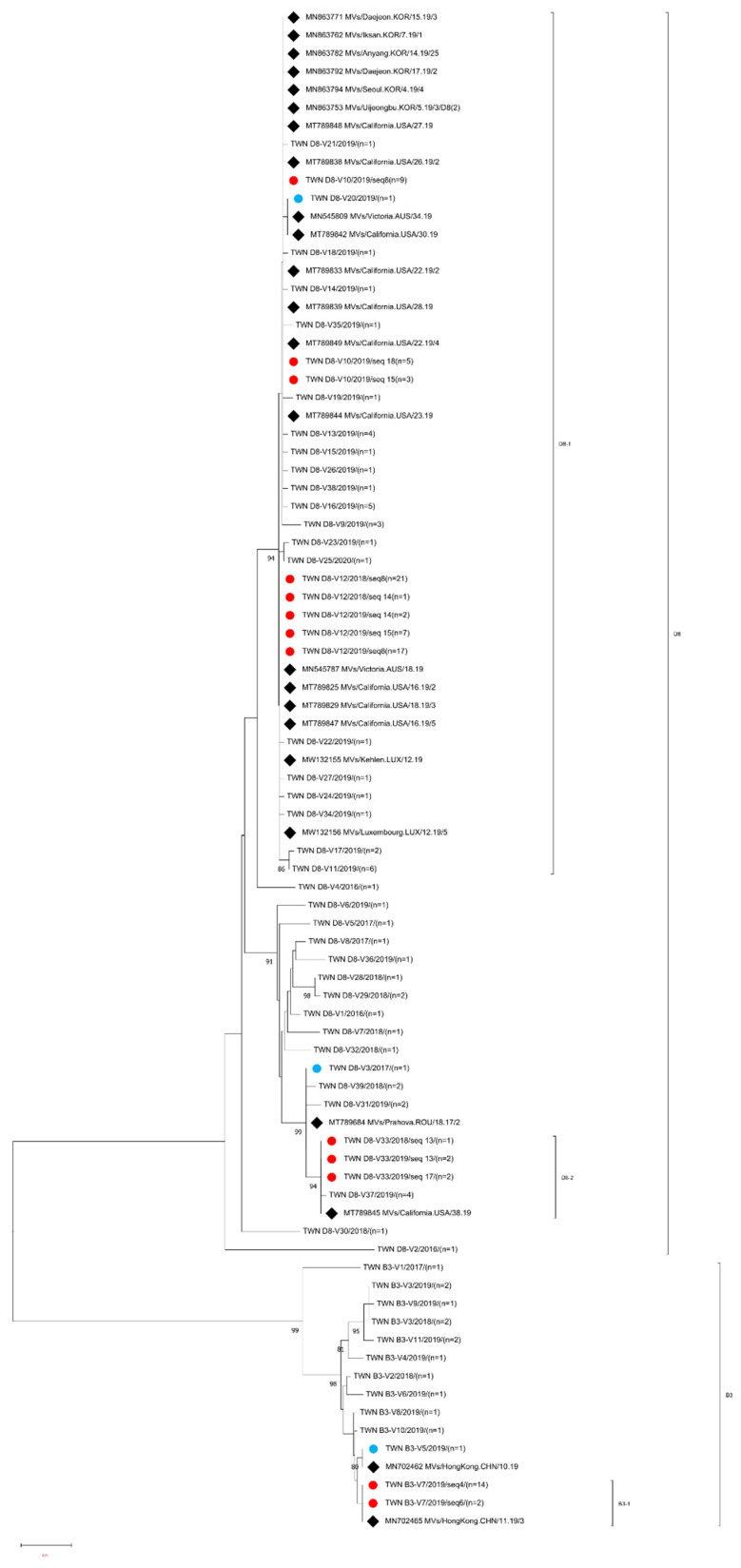
Phylogenetic tree of the genotype B3 and D8 MeV strains. The phylogenetic tree was constructed using the neighbor-joining method based on the MF-NCR sequences. Circles colored in blue indicate strains detected in Taiwan that had MF-MCR sequences identical to those of the MeV strain found in GenBank. The circle colored in red indicates that strains detected in Taiwan had identical MF-MCRs but deviated from N-450 sequences. Diamonds colored in black indicate the overseas MeV strains with an MF-NCR sequence identical to one or another of the MeV strains detected in Taiwan. The number (n) of each MeV strain with an identical MF-NCR sequence is shown. The other sequences without additional labels were other B3 and D8 MF-NCR variants. Genotype D8 sub-lineage D8-1, D8-2 and genotype B3-1 as tentatively proposed in this study are indicated.

**Table 1 viruses-15-00211-t001:** The detail sequences of primer/probe used for real-time RT-PCR, N-450 genotyping and MF NCR sequence window.

Primer ID	Sequence 5′-3′	Note
1905F	gcagcatggtcagaaatatcag	for real-time RT-PCR
2030R	gcacYgccttcagYtgatcc	
1967P	ttgctgagacccgaactgcctgcct	
MV59	gatatgtgacattgatacatatat	for N-450 genotyping RT-PCR
MV64	tataacaatgatggagggtag	
Me 214	taacaatgatggagggtagg	for N-450 genotyping nested PCR
Me 216	tggagctatgccatgggagt	
4200F	ggcaccagtcttcacattagaag	for segment 1 MF NCR RT-PCR/heminested PCR
4212F	cacattagaagYacaggcaa	
4869R	cttggccctRagttttgtttag	
4801F	cacaagcgaccgaggtgac	for segment 2 MF NCR RT-PCR/heminested PCR
4811F	acccaaccRcaggcatccga	
5609R	cgagtcataactttgtagcctgc	

**Table 2 viruses-15-00211-t002:** The number of reported measles cases and genotype data based on the N-450 and MF NCR sequence window in Taiwan, 2015–2020.

Year	The Number of Reported Measles Cases	The Number of Laboratory Confirmed Measles Cases ^a^	The Number of Cases MV Genotype Data						
			Total	Genotype (N-450)			Total	Genotype (M/F NCR-1018)	
				H1	D8	B3		D8	B3
2015	141	29	27	27	0	0	0	0	0
2016	114	14	13	7	6	0	3	3	0
2017	99	6	6	1	4	1	4	3	1
2018	471	40	37	0	34	3	34	31	3
2019	982	140	135	0	101	34	115	89	26
2020	172	1	1	0	1	1	1	1	0
Total	1979	230	219	35	146	38	157	127	30

^a^ Laboratory confirmed cases should met at least one of the criteria of serology measles IgM positive or throat swab/urine RT-rPCR positive.

**Table 3 viruses-15-00211-t003:** N-450 variant identified in Taiwan, 2015–2020.

Genotype	N-450 Variant ID	Number	WHO Named Strain/Accession Number * Accession Number **	Epidemiologic Link ***
H1	H1-seq 1	24	MVs/HongKong.CHN-49.12[H1]/KC417295	A(5)-CHN; B(17);C(2)
	H1-seq 2	1		C(1)
	H1-seq 3	2	MVs/HongKong.CHN-42.11[H1]/JQ031212	A(1)-CHN; B(1)
	H1-seq 4	1		A(1)-CHN
	H1-seq 5	1	AB968373, LC002658	A(1)-VNM
	H1-seq 6	1		A(1)-CHN(HongKong)
	H1-seq 7	4	KY056247	A(2)-CHN; B(2)
	H1-seq 8	1	MH979988	A(1)-CHN
B3	B3-seq 1	1	MVi/Gombak.MYS/44.16[B3]/KU678417	C(1)
	B3-seq 2	8	MVi/Gombak.MYS/40.15[B3]/KU714612	A(6)-PHL; B(1);C(1)
	B3-seq 3	1	MK633031	A(1)-PHL
	B3-seq 4	25	MVi/Marikina City.PHL/10.18[B3]/MN602382	A(6)-PHL,CHN,JPN,Europe ****; B(18);C(1)
	B3-seq 5	1	MT789788, 789389, 789790, MT386953	A(1)-USA/CHN
	B3-seq 6	2		B(1);C(1)
D8	D8-seq 1	1	KX377943	C(1)
	D8-seq 2	1		A(1)-IND
	D8-seq 3	4	MVs/Osaka,JPN/29.15[D8]/LC072667	A(3)-JPN, THA;B(1)
	D8-seq 4	1	MVs/Victoria.AUS/6.11[D8]/KF469368	A(1)-IND/CHN/USA/SGP
	D8-seq 5	1	MF092792	A(1)-IDN
	D8-seq 6	3	MVi/Hulu Langat.MYS/26.11[D8]/JX486001	A(1)-FRA/BEL/NLD
	D8-seq 7	1		A(1)-THA
	D8-seq 8	90	MVs/Gir Somnath.IND/42.16[D8]/KY120864	A(35)-THA, VNM,CHN(Macao), KOR, IDN, JPN, BEL, PHL,NZL; B(43);C(12)
	D8-seq 9	3		A(2)-IDN;B(1)
	D8-seq 10	1	MG652493	C(1)
	D8-seq 11	4	MVs/Samut.Sakhon.THA/49.16[D8]/MK079566	A(4)-Thailand
	D8-seq 12	1	MVs/Herborn.DEU/05.17[D8]/KY973620, T789841, KY973602	A(1)-GBR
	D8-seq 13	6	MVs/Samut Sakhon.THA/8.18[D8]/MN602383	A(3)-THA, COL
	D8-seq 14	4	MVs/Dagon Seikkan.MMR/5.18[D8]/MN602384	A(3)-THA,MMR;B(1)
	D8-seq 15	12		A(3)-CHN, JPN;B(9)
	D8-seq 16	1		A(1)-IDN
	D8-seq 17	6	MT555111, LC521318, MT789845	A(3)-THA; B(2); C(1)
	D8-seq 18	6		A(4)-THA, COL; B(2)

* N-450 variant identical to WHO named strain sequence. ** N-450 variant not classified to WHO named strain but matched to sequences deposited in Genbank. *** A: Importation was defined as cases with international travel histories in 3 weeks, the case numbers from different categories of epidemiological link were shown in parentheses and the countries the imported cases had visited were indicated with a 3-letter country ISO code. B: Domestic transmission was defined as cases that either had an epidemiological linkage to imported cases or had histories of contact with each other. C: Domestic case was defined as sporadic case with unknown source of infection. **** Three cases travelled in more than one European country including SWE, NOR, DNK, FIN, ISL and GBR. The exact travel history from each sequence can be found in the Appendix A.

**Table 4 viruses-15-00211-t004:** The relative N-450 and MF-NCR variants detected in 26 epidemiological link (Epi-link) cases.

Epi-Link ID	Genotype	Case Number	N-450 Variant	MF NCR Variant	Imported from
1 ^a^	H1	16	H1-seq1/seq2	NA ^e^	NA
2	H1	2	H1-seq1	NA	NA
3	H1	2	H1-seq7	NA	NA
4	D8	2	D8-seq3	NA	Thailand
5 ^b^	D8	21	D8-seq8	D8-V12	Thailand
6	D8	2	D8-seq9	D8-V29	Indonesia
7	B3	2	B3-seq2	B3-V2	the Philippines
8	D8	3	D8-seq8	D8-V9	Vietnam
9	D8	6	D8-seq8	D8-V11/V17	Vietnam
10	D8	2	D8-seq8	D8-V12	NA
11	D8	2	D8-seq8	D8-V12	Japan
12	D8	2	D8-seq14	D8-V12/V34	Myanmar
13 ^c^	D8	10	D8-seq15	D8-V10/V12/V35	China
14	D8	4	D8-seq8	D8-V13	Thailand
15 ^d^	B3	18	B3-seq4/seq6	B3-V7/V8	Europe ^f^
16	D8	5	D8-seq13	D8-V33	Thailand
17	D8	2	D8-seq15	D8-V12	Japan
18	B3	2	B3-seq4	B3-V7/V10	NA
19	D8	2	D8-seq8	D8-V10	NA
20	B3	3	B3-seq4	B3-V7	NA
21	D8	2	D8-seq8	D8-V10	Vietnam
22	D8	2	D8-seq8	D8-V22	Vietnam
23 ^c^	D8	10	D8-seq8	D8-V16	Vietnam
24	D8	3	D8-seq18	D8-V10	Cambodia
25	D8	3	D8-seq17	D8-V37	Thailand
26	B3	2	B3-seq2	B3-V11	Italy

^a^ Epi-link 1 was linked to a measles transmission cluster in a tax-free shop. ^b^ Epi-link 5 was linked to a measles transmission cluster at an international airport. ^c^ Epi-link 13 and 23 were linked to a measles transmission cluster in two hospitals. ^d^ Epi-link 15 was linked to measles transmission in a hospital and a science and technology park. ^e^ NA: not available. ^f^ The index case of Epi-link 15 had a travel history to European countries included Sweden, Denmark, Finland, Norway and Iceland.

## Data Availability

The datasets supporting this study are included within the article and its tables, figures, and Appendix A. Additional data may be available from the corresponding authors upon request.

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
