# Peer review of "Genetic Characteristics of Measles Viruses Isolated in Taiwan between 2015 and 2020"

_viruses, 2023, doi:10.3390/v15010211_

Round 1
Reviewer 1 Report
Lines 71-72 " The RT- qPCR result was considered positive if there was amplification within 40 cycles." : it is advisable to state the cutoff of the RT-PCR (ie. up to which cycle a result is considered to be positive)
Lines 78-79 "When throat and urine samples were confirmed positively for MeV by qRT-PCR assay, the samples were inoculated into Vero/hSLAM cells for further isolation of the MeVs" : more detail is needed on the cell culture methods.
Author Response
- Lines 71-72 " The RT- qPCR result was considered positive if there was amplification within 40 cycles.": it is advisable to state the cutoff of the RT-PCR (ie. up to which cycle a result is considered to be positive)
Response: We have modified the sentence as “The RT-rPCR result was defined as positive if the cycle threshold value of RT-rPCR was less 40.”(Line 77-78)
- Lines 78-79 " When throat and urine samples were confirmed positively for MeV by qRT-PCR assay, the samples were inoculated into Vero/hSLAM cells for further isolation of the MeVs."more detail is needed on the cell culture methods.
Response: We have added the detailed methods and references as “The samples were inoculated into Vero/hSLAM cells-a cell line developed by Dr.Yangi and colleagues at Kyushu University, Fukuoka, Japan, for further isolation of the MeVs [11]. The culture medium was supplemented with Geneticin to retain SLAM (signaling lymphocyte-activation molecule) expression [12].” (Line 84-87)
Added two new references.
[11] Ono, N.; Tatsuo, H.; Hidaka, Y.; Aoki, T.; Minagawa, H.; Yanagi, Y., Measles viruses on throat swabs from measles patients use signaling lymphocytic activation molecule (CDw150) but not CD46 as a cellular receptor. J Virol 2001, 75, (9), 4399-401.
[12]Manual for the laboratory diagnosis of measles and rubella virus infection (second edition). WHO/IVB/07.01.

Reviewer 2 Report
The manuscript entitled: “Genetic Characteristics of Measles Viruses isolated in Taiwan Between 2015 and 2020” by Cheng et al, submitted to the journal Viruses (manuscript ID 2126462) has been reviewed. The manuscript provides a study on the genotype distribution of measles virus in aforementioned years. It is well written although it may benefit from some minor editing, e.g. use of “blasted” (line 257) for a Blast sequence comparison. Although the data may not be very interesting for a more general audience, it does provide an overview of the genetic variants of measles found on Taiwan and the elimination of an endemic genotype H1. Although the authors extensively quote WHO policies on verification of measles elimination, they fail to conclude if measles has been eliminated based on the data they present. Furthermore, it would be very interesting if the authors draw a conclusion on the use of MF sequencing and the increased resolution they claim it has. Authors are encouraged to discuss this.
In the introduction the authors spend a great deal of text on measles elimination and verification criteria. Please condense and refer to relevant papers. What is more important is to provide more epidemiological background on measles in Taiwan during the given period: incidence, vaccination coverage, case distribution (age and geography).
In the Materials and Methods section the authors do not make it clear if the sequencing primers they have used have been developed in-house or have been published previously.
Authors should use RT-rPCR rather than qRT-PCR given that no quantitative results of the PCR assay they used have been given.
Table 2: Number of laboratory (not laboratories) confirmed
Please clarify what you mean with “domestic transmission” and “domestic case”. Does that mean measles is endemic and surveillance has not been up to par to detect these transmission chains?
Can the authors add an extra column to the Table 3 indicating epidemiologic link?
Line 250: please note that given the length of the fragment of 450 nucleotides used for measles genotyping and the low genetic variability, the use of N450 for molecular epidemiological analysis is not recommended but should rather be used to confirm or rule out epidemiological linkage.
Line 404 onwards: please use phylogenetic tree rather than dendrogram.
The authors fail to conclude if MF sequencing adds to increased resolution and aids in defining chains of transmission. A deeper analysis of this data and particularly the comparison with N450 should be presented.
Author Response
We want to thank the editor and reviewers for your efforts in reviewing our manuscript. You did read the manuscript carefully and gave us good suggestions. We appreciate the comments and suggestions and think that they are all very helpful. The point-to-point responses to the comments are replied as follows.
- It is well written although it may benefit from some minor editing, e.g. use of “blasted” (line 257) for a Blast sequence comparison.
Response: We have modified the sentence as “Among the 11 MF NCR variants, 2 variants (B3-V5 and B3-V7) could match with identical sequences deposited in GenBank.” (Line 271)
- Although the authors extensively quote WHO policies on verification of measles elimination, they fail to conclude if measles has been eliminated based on the data they present.
Response: We have added the conclusion in the revised manuscript as “The data of genotypes according to two sequence windows provide evidences that measles elimination has been reached in Taiwan”.(Line 442-444)
- In the introduction the authors spend a great deal of text on measles elimination and verification criteria. Please condense and refer to relevant papers. What is more important is to provide more epidemiological background on measles in Taiwan during the given period: incidence, vaccination coverage, case distribution (age and geography).
Response: We have condensed and referred to relevant papers in the introduction section and added more epidemiological background on measles in Taiwan as” In Taiwan, the measles cases decreased since 2015, and reached lowest in 2017 with only 6 cases were confirmed, but it rebounded to 140 cases in 2019 as the worldwide trend [4]. The vaccine coverage rate of the first dose measles, mumps and rubella (MMR) that given to 12 months old children reached over 98 % between 2015 and 2020, and the second dose MMR that given to children before entrance to elementary school improved from 93.42 % in 2015 to 96.77 % in 2020. Among 230 measles cases detected between 2015 and 2020, 7.8 % were from children under 1-year-old, 3.9 % at children age 1-6 years old, 1.3 % at school age children (7-17 years old), 77.8 % at adults age 18-40 years old, and 9.2 % at adults age over 40-year-old. Nearly 74 % cases were from northern Taiwan, and that from the central and southern Taiwan accounted for 15.5 % and 10.5 %, respectively”. (Line 38-48) and also update supplementary table by adding extra column of age.
- In the Materials and Methods section the authors do not make it clear if the sequencing primers they have used have been developed in-house or have been published previously.
Response: Revised version Line 104-106. “The forward and reverse sequencing primers used were Me 216 and Me 214 for N-450 and 4212F/4869R, 4811F/5609R for segment 1 and segment 2 of MF-NCR, respectively.”
- Authors should use RT-rPCR rather than qRT-PCR given that no quantitative results of the PCR assay they used have been given.
Response: We have changed “qRT-PCR” to “RT-rPCR” throughout the revised paper.
(Line 72, 73, 83, 88, 121, 122 and 130)
- Table 2: Number of laboratory (not laboratories) confirmed
Response: We have modified as “laboratory” in revised Table 2.
- Please clarify what you mean with “domestic transmission” and “domestic case”. Does that mean measles is endemic and surveillance has not been up to par to detect these transmission chains?
Response: We have clarified “domestic transmission” and “domestic case” in footnotes of Table 3 as” Domestic transmission defined as cases were either had epidemiological linkage to imported case or had contact histories each other” and “Domestic case defined as sporadic case with untraceable source of infection.”(Line 250-252)
- Can the authors add an extra column to the Table 3 indicating epidemiologic link?
Response: We have modified Table 3 as reviewer’s suggestion.
- Line 250: please note that given the length of the fragment of 450 nucleotides used for measles genotyping and the low genetic variability, the use of N450 for molecular epidemiological analysis is not recommended but should rather be used to confirm or rule out epidemiological linkage.
Response: We have modified the statement to “When multiple importations with identical N-450 sequences made it difficult to clarify the transmission chain, the use of N450 for molecular epidemiological analysis was not recommended but should rather be used to confirm or rule out epidemiological linkage.” as reviewer’s suggestion.(Line 263-266)
- Line 404 onwards: please use phylogenetic tree rather than dendrogram.
Response: We have replaced dendrogram to phylogenetic tree throughout the revised paper.
- The authors fail to conclude if MF sequencing adds to increased resolution and aids in defining chains of transmission. A deeper analysis of this data and particularly the comparison with N450 should be presented.
Response: We added statement of MF-NCR sequencing in the conclusion section as “A new sequence window of MF-NCR offers a higher resolution compared to N-450 for genotype D8 MeVs (39 variants vs 18 variants) and genotype B3 MeVs (11 variants vs 6 variants). It’s highly recommended to apply MF-NCR sequencing as an extra tool for molecular epidemiology and defining chains of transmission. Because same MeV variants that were classified according to the N-450 sequence window might have different MF-NCR sequences, and vice versa, a deeper analysis of N-450, MF-NCR sequencing and whole genome combined with epidemic investigation is ongoing.(Line 440-446)

Reviewer 3 Report
The authors present a very thorough analysis of the molecular epidemiology of measles in Taiwan from 2015 to 2020. Three genotypes H1, B and D8 were detected, but after 207, only genotypes D8 and B3 were detected. This is consistent with the global pattern. Because many of the viruses had identical N-450 sequences, the authors used extended sequencing of the MF-NCR to provide greater resolution to the molecular data. The combination of MF-NCR sequences and standard case tracing and epidemiological investigations was able to resolve most of the transmission pathways and suggest that no strain was continually transmitted from greater than 12 months. While this study provides an excellent example of the value of using extended sequencing windows, the authors should consider the comments listed below.
1. While the paper is well written, it would benefit by a thorough editing to correct the errors with English Grammar.
2. Line 44, please update references. Cite the following instead of reference #4. https://pubmed.ncbi.nlm.nih.gov/36417303/; https://apps.who.int/iris/bitstream/handle/10665/363330/WER9739-eng-fre.pdf?sequence=1&isAllowed=y
3. Line 36, Please include that verification of elimination requires evidence that transmission was interrupted for 36 months.
4. Line 28. Please re-write sentence because there were no vaccine preventable diseases in the pre vaccine era.
5. The authors use the terms “dominant” and “replacement” to describe the circulation of measles lineages. However, there is no real evidence that any genotype is dominant and has the ability to replace another genotype. Rather, the genotypes detected reflect the distribution of susceptible individuals, exposure, and travel. The preferred language would be that genotype B3 and D8 “were the most frequently detected genotype”.
6. The main comment is that the paper is very thorough but very difficult to follow. All the transmission chains are described in intricate detail. It may be better to focus on the longer chains and the resolution of these chains by MF-NCR sequencing than to describe everything. Also, the graphics and tables are quite detailed, but don’t give the reader a sense of time. The authors may wish to consider displaying the detection of sequence variants as an epidemiologic cure (i.e., show detection of variants by epidemic week for each year). This would allow the reader to easily see the length of transmission chains.
